# Ceramide/Sphingosine 1-Phosphate Axis as a Key Target for Diagnosis and Treatment in Alzheimer’s Disease and Other Neurodegenerative Diseases

**DOI:** 10.3390/ijms23158082

**Published:** 2022-07-22

**Authors:** Antía Custodia, Daniel Romaus-Sanjurjo, Marta Aramburu-Núñez, Diego Álvarez-Rafael, Laura Vázquez-Vázquez, Javier Camino-Castiñeiras, Yago Leira, Juan Manuel Pías-Peleteiro, José Manuel Aldrey, Tomás Sobrino, Alberto Ouro

**Affiliations:** Neuro Aging Laboratory Group (NEURAL), Clinical Neurosciences Research Laboratories (LINCs), Health Research Institute of Santiago de Compostela (IDIS), 15706 Santiago de Compostela, Spain; antia.custodia.malvido@sergas.es (A.C.); daniel.romaus.sanjurjo@sergas.es (D.R.-S.); marta.aramburu.nunez@sergas.es (M.A.-N.); diegoarsi08@gmail.com (D.Á.-R.); laura.vazquez.vazquez@sergas.es (L.V.-V.); anxacc@gmail.com (J.C.-C.); yagoleira@gmail.com (Y.L.); juan.manuel.pias.peleteiro@sergas.es (J.M.P.-P.); jaldreyv@yahoo.com (J.M.A.)

**Keywords:** Alzheimer’s disease, β-amyloid, ceramide, metabolism, sphingosine 1-phosphate, sphingolipids, sphingolipidomics, tau

## Abstract

Alzheimer’s disease (AD) is considered the most prevalent neurodegenerative disease and the leading cause of dementia worldwide. Sphingolipids, such as ceramide or sphingosine 1-phosphate, are bioactive molecules implicated in structural and signaling functions. Metabolic dysfunction in the highly conserved pathways to produce sphingolipids may lead to or be a consequence of an underlying disease. Recent studies on transcriptomics and sphingolipidomics have observed alterations in sphingolipid metabolism of both enzymes and metabolites involved in their synthesis in several neurodegenerative diseases, including AD. In this review, we highlight the most relevant findings related to ceramide and neurodegeneration, with a special focus on AD.

## 1. Introduction: Alzheimer’s Disease and Sphingolipids

Alzheimer’s disease (AD) is considered the most prevalent neurodegenerative disease and the leading cause of dementia worldwide [1]. Based on clinical symptoms, the disease is characterized by memory loss (especially of recently learned information), temporal disorientation, language and personality disturbances, aggressiveness, agitation, and hallucinations, among others [2]. According to the *World Health Organization* (WHO), 55 million people suffer from dementia worldwide, with approximately 10 million new cases diagnosed each year [1,3]. In 2019, approximately 1.62 million people died from dementia, making it the seventh cause of death, and the fourth one in people over 70 years of age [4]. In addition, it is estimated that AD involves an annual cost of USD 1 trillion worldwide, which, together with the exponential increase in the number of cases due to the aging of the population, represents a great burden on healthcare systems [1].

The major neuropathological hallmarks of AD are the extracellular deposits of β-amyloid (Aβ) forming amyloid plaques as well as the intracellular accumulation of hyperphosphorylated tau protein in neurofibrillary tangles (NFT) [5]. In addition, AD is also characterized by vascular alterations [6], neuroinflammation [7], oxidative stress [8], and alterations in lipid and sphingolipid (SL) metabolism [9], among others.

The involvement of lipid metabolism in AD progression is well established. In this regard, the main genetic risk factor for sporadic late onset-AD is the presence of the ε4 allele of the apolipoprotein E gene (*APOE4*) [10]. There are different isoforms of *APOE*; *APOE3* (which does not affect AD) is the most abundant, *APOE2* (which appears to be beneficial for AD) is the least abundant; and *APOE4* which is linked to the onset of AD by increasing the risk of developing the disease. Interestingly, ApoE is highly expressed in the brain. ApoE plays a critical role in cholesterol and other lipids’ transport and redistribution [11,12]. In addition, ApoE is also involved in other cellular processes such as cytoskeleton assembly and stability, mitochondrial function and integrity, and dendrite function and morphology [13]. Specifically, ApoE4 associates with Aβ more rapidly and stably than the other ApoE isoforms, facilitating the formation of amyloid plaques. Interestingly, ApoE4 mediates other pathological functions related to AD; for example, it stimulates tau phosphorylation, inhibits neurite outgrowth, and impairs neuronal plasticity and the integrity of the blood-brain barrier (BBB), among others [13]. Moreover, there are different AD-related single nucleotide polymorphisms (SNPs) related to plasma lipid levels. For example, FERM Domain Containing Kindlin 2 (*FERMT2)* and Membrane Spanning 4-Domains A6A (*MS4A6A)* show a significant differential association between AD patients and controls in all lipid classes (the association of *MS4A6A* with phosphatidylinositol (PI) levels is noteworthy); ATP Binding Cassette Subfamily A Member 7 (*ABCA7)* is associated with diglycerides (DG) and PI levels; Clusterin (*CLU)* with sphingomyelin (SM) and phosphatidylethanolamines (PE) levels; PI binding clathrin assembly protein *(PICALM)*, Solute carrier family 24 member 4 (*SLC24A4)* and Sortilin-related receptor 1 (*SORL1)* with cholesteryl esters (ChE) levels; Bridging integrator 1 (*BIN1)* with triglycerides (TG) levels; and Complement C3b/C4b Receptor 1 (*CR1)* with PE levels [14]. Thus, the clear involvement of different types of lipids in AD is evident.

SLs are bioactive lipids with structural and second messenger functions. SLs participate in different cellular processes such as cell proliferation, migration, apoptosis, autophagy, senescence, and inflammation [15,16,17,18,19,20]. Moreover, SLs are highly expressed in the central nervous system (CNS), being the main component of the plasma membrane and myelin [21]. In addition, SLs have a strong relationship with different neurodegenerative diseases [9] and brain injury [22]. Among all, we highlight ceramide (Cer), the central metabolite in SL metabolism, and sphingosine 1-phosphate (S1P), the most studied. It is well known that Cer is a proapoptotic molecule, as it stimulates inflammation, autophagy, and oxidative stress by mitochondrial dysfunction [9,23], while its phosphorylated analog (ceramide 1-phosphate, C1P) or S1P stimulates cell survival, proliferation, and migration, among other functions [15]. S1P is also involved in neurodevelopment, synaptic transmission, neuroinflammation, and autophagy [9].

In this regard, genetic alterations in SL metabolism have been observed in brain tissue, cerebrospinal fluid (CSF), and the blood of patients with neurodegeneration [24,25,26,27]. Moreover, SLs have been associated with different pathological processes such as neuroinflammation and amyloid plaque formation, and thus are being proposed as promising biomarkers [28,29,30]. Consistently, mass spectrometry analysis of SL, called sphingolipidomics, has been performed to study the SL profile in brain samples as well as in biological fluids such as CSF, serum, or plasma. In fact, several studies have observed SL alterations in AD patients and patients with other neurodegenerative diseases compared to controls, and even in early clinical stages such as mild cognitive impairment (MCI) [31]. Moreover, these variations have been detected in different stages of AD progression, from prodromal to more advanced stages [31]. Therefore, SLs could be considered a potential biomarker [31], an aspect that will be discussed in this review. This review encompasses the most recent studies on the involvement of sphingolipids in neurodegenerative diseases, and their use both as therapeutic targets and biomarkers. In particular, we will focus on the role of SL metabolism in AD.

## 2. Sphingolipid Metabolism

Cer is considered the central metabolite of SL metabolism. So far, there are more than 30 described enzymes implicated in Cer metabolism. It is important to highlight that these enzymes are highly conserved and strictly regulated. The metabolism of SL can be differentiated into three main pathways, depending on the origin of the metabolites and their cellular location (Figure 1).

### 2.1. The De Novo Pathway

The de novo pathway takes place in the endoplasmic reticulum (ER). Serine palmitoyltransferase (SPT) is the first enzyme in the de novo pathway, which condensates palmitate and serine to produce 3-ketodehydrosphingosine (known also as 3-ketosphinganine). Being the first enzyme, it is of vital importance in controlling the rate of SL synthesis by de novo pathway. SPTLC1, SPTLC2, ssSPTa/b, and a negative regulatory subunit ORMDLs (homologs of the yeast and plant Orms) are the subunits that form the SPT complex in the ER membranes [32,33]. Subsequently, the 3-keto-dihydrosphingosine reductase (KDR) incorporates a hydrogen atom to 3-ketodehydrosphingosine, to synthesize sphinganine (Spha). Then, the ceramide synthase (CerS) enzyme introduces another acyl-CoA to form dihydroceramide (dhCer). So far, six isoforms of CerS (CerS1–6) have been described in mammals and plants [34]. Each isoform has a different affinity for Acyl-CoA molecules depending on the length of the chain. In this regard, CerS1 has more affinity to produce 18 carbon chain Cer (C_18_-Cer), CerS2 to C_22/24_-Cer, CerS3 to C_26_-Cer, CerS4 to C_18/20_-Cer, CerS5 to C_14/16_-Cer, and CerS6 to C_14/16_-Cer. Interestingly, C_16_-Cer synthesized by CerS6 is related to pro-survival events, while C_18_-Cer and longer are involved in pro-apoptotic pathways. Remarkably, CerS1 is highly expressed in the CNS. Finally, dihydroceramide desaturase (DEGS) introduces a double bond in position 4–5 trans of dhCer to form Cer.

### 2.2. The Sphingomyelinase (SMase Pathway)

SMase is a hydrolytic enzyme that degrades sphingomyelin (SM) at membrane level to generate Cer and phosphocholine in the cytosol, lysosomes, or lumen. So far, five isoforms have been described and classified according to their location, ionic regulation, and optimal pH activity. Acid SMase (aSMase) [35] is found in the lysosomes and plasma membrane [35]; its secreted form (Zn^2+^-dependent aSMase) is observed in saliva and serum [36]; neutral SMase (nSMase) has been detected in the nucleus, ER and plasma membrane [37]; and alkaline SMase (alkSMase) is secreted to intestinal tract lumen and human bile [38,39]. Oppositely, sphingomyelin synthase (SMS) catalyzes the production of SM from Cer. To date, three different SMSs have been described; SMS1, SMS2, and SMS-related protein (SMSr) [40]. It should be noted that different noxious stimuli can activate acid and neutral SMase and increase Cer synthesis, including different cytokines and interleukins, radiation, and cellular signals of cell cycle arrest or apoptosis [41].

### 2.3. The Salvage Pathway

The salvage pathway involves catabolic reactions to degrade complex SLs in lysosomes. Complex SLs are degraded through different reactions to produce Lactosyl-ceramide (LacCer). Then, LacCer hydrolase catalyzes the hydrolysis of LacCer to Glucosyl-Ceramide (GlcCer). Subsequently, GlcCer is hydrolyzed by acid β-glucosidase 1 (β-GCase) to form Cer. β-GCase is encoded by the GBA1 gene, whose deficiency or dysfunction produces GlcCer accumulation, leading to the development of the lysosomal storage disease known as Gaucher’s disease. In addition, mutations in *GBA1* are associated with Parkinson’s Disease (PD) [42,43,44]. In the opposite direction, the glucosylceramide synthase (GCS) enzyme converts Cer into GlcCer. Cer can be transformed to sphingosine (Sph) by the ceramidase activity. Ceramidases are a small group of enzymes that differ in their optimal pH activity. Thus far, there are three alkaline enzymes, known as ACER1, ACER2, and ACER3, an acidic ceramidase (ASAH1), and a neutral ceramidase (ASAH2) [45,46]. Importantly, ASAH1 is ubiquitously found in lysosomal compartments. However, ASAH2 is observed in plasma membranes, being mainly expressed in the colon and small intestine [45]. Finally, Sph is secreted to the cytosol to produce Cer (by CerS in ER) or S1P.

### 2.4. Ceramide Kinase/Ceramide 1-Phosphate Phosphatase (CerK/CPP) and Sphingosine Kinase/Sphingosine 1-Phosphate Phosphatase (SphK/SPP) Axis

As explained, Sph can be phosphorylated to form S1P by sphingosine kinase (SK) activity. S1P is involved in inflammatory processes through its interaction with specific membrane receptors (S1PR), such as glia activation and vascular-mediated inflammation, among others [47,48,49]. Five S1P receptors (S1PR1–5) have been described [50]. Meanwhile, S1P phosphatase (SPP) or lipid phosphate phosphatase (LPP) catalyzes the dephosphorylation of S1P to Sph [19]. In addition, S1P can be catalyzed by the activity of S1P lyase to produce phosphoethanolamine and hexadecanal [51].

In addition, Cer can undergo phosphorylation in the Golgi apparatus by ceramide kinase (CerK) to form ceramide 1-phosphate (C1P) [52]. C1P is implicated in functions such as proliferation/survival, migration, metabolism, and inflammation, among others [15].

## 3. Neurodegeneration and Sphingolipid Metabolism

As shown above, there are different species of SLs, and their balance is of vital importance for cell fate. On this matter, a very high ratio for non-phosphorylated forms (Cer/C1P-Sph/S1P) leads to cell death and dysfunction, whereas the opposite ratio gives rise to cell proliferation, cell survival, and inflammatory processes, depending on the cell type and ratio [15,17,18,53]. Remarkably, the unbalance of SLs species as a result of the dysregulation of SLs metabolism is a common event in several neurodegenerative diseases, such as AD, PD, or multiple sclerosis (MS).

It is well known that Cer is a pro-apoptotic and pro-inflammatory molecule. For example, Cer leads to neuronal apoptosis through interaction with the phosphatidylinositol 3-kinase/protein kinase B (PI3K/Akt) pathway, Bcl-2 antagonist of cell death (BAD), glycogen synthase kinase 3-β (GSK3-β), Forkhead family transcription factors (FOX proteins), mitochondrial depolarization (promoting its permeabilization), and caspase-3 activation [54]. Mitochondria plays an important role in the activation of apoptosis in mammalian cells. Herein, Cer interacts with the pro-apoptotic protein Bcl-2-associated X protein (BAX) to form pores in the outer mitochondrial membrane (OMM), increasing its permeability (Figure 2). This event leads to a release of pro-apoptotic molecules (in the cytosol), such as cytochrome C, mitochondrial serine proteinase HtrA2 (also called Omi), second mitochondrion-derived activator of caspase (Smac/DIABLO) or apoptosis-inducing factor (AIF) [55,56]. In addition, exogenous treatment with a short analog of Cer (C_2_-Cer) induces the activation of caspase-3 and -9 in neuronal primary culture, cerebellar granule cells, and the neuroblastoma cell line SH-SY5Y [54,57,58]. Moreover, C_2_-Cer treatment also induces neuronal death by inactivation of PI3K/Akt and extracellular signal-regulated kinases (ERK) pathways, in addition to B-cell lymphoma 2 (Bcl-2), as well as activation of c-Jun N-Terminal Kinase (JNK) and GSK3-β pathways, BAX, Harakiri (Hrk), and poly ADP-ribose polymerase 1 (PARP-1) [59,60]. Interestingly, GSK3-β is one of the kinases responsible for Tau hyperphosphorylation and has even been linked to Aβ formation [61]. In this sense, recent studies have demonstrated that overexpression of CerS or ablation of ASAH1 stimulates or confers resistance to apoptosis, respectively [59,60].

As described, cellular stress is involved in the onset and progression of neurodegenerative diseases. Cellular stress may be due to a wide range of environmental factors, such as temperature, exposure to toxins or UV radiation. Interestingly, these factors can activate JNK signaling leading to activation of nSMase1, with an increase in Cer levels and subsequent initiation of apoptosis [41]. Moreover, noxious stimuli such as different cytokines and interleukins, a cellular signal of cell cycle arrest, or apoptosis can activate SMase and increase Cer as well [61]. Protein phosphatase 2A (PP2A) is a member of serine/threonine protein phosphatase whose imbalance has been described in several neurodegenerative diseases. Of interest, PP2A was observed as a modulator of JNK, ERK and p38 pathways, promoting the production and release of a pro-inflammatory cascade in lung epithelial cells [41]. In addition, TNF-α increases intracellular Cer levels, an event that promotes PP2A activity [41]. Furthermore, it was demonstrated that Cer stimulates cell cycle arrest and even apoptosis by a PP2A-dependent Akt dephosphorylation mechanism and a subsequent p27 increase in PC-3 cancer prostate cancer cell line [62].

It is well established that inflammatory responses mediated by microglia and astrocytes exert protective functions in AD. However, dysregulation of these mechanisms contributes to disease progression [63]. Elevated intracellular Cer levels in astrocytes were associated with increased neuroinflammation in frontotemporal lobar dementia or Pick’s disease [64]. Moreover, oxidative stress due to an elevation of reactive oxygen species (ROS) plays a key role in the development of neurodegenerative diseases. Nitric oxide (NO) is an important mediator of several functions, such as inflammation or vasodilation; and plays a role in neurotoxicity associated with several neurological disorders, including stroke, AD, PD, and HIV dementia [65]. Furthermore, both SMase and C_2_-Cer treatment activates the pro-inflammatory enzyme inducible nitric oxide synthase (iNOS) by a mechanism dependent on a nuclear factor kappa-light-chain-enhancer of activated B cells (NF-κB) [66]. Interestingly, both C_2_-Cer and Fingolimod (S1PR antagonist) treatment induce an increase in interleukin (IL)-1β and caspase-1 levels in microglial culture cells leading to activation of the inflammasome [67]. Recently, it was also shown that the lack of *SMS2* also suppressed the activation of microglia through the inhibition of NF-𝜅B and reduced brain ischemic injury [68]. The pro-inflammatory cytokine, tumor necrosis factor α (TNF-α) has a central role in microglial activation, and mediates neuronal and synapses loss. Interestingly, the exposure to TNF-α activates SMase with subsequent elevation of Cer levels [69].

It is well known that autophagy is a fundamental process to eliminate protein aggregations such as Aβ and tau in AD; α-synuclein (α-syn) in PD; huntingtin in Huntington’s disease (HD); or TAR DNA-binding protein 43 (TDP-43) in amyotrophic lateral sclerosis (ALS) [70]. Interestingly, Henriques and co-workers demonstrated a SL metabolism alteration in ALS mice models, with increased levels of glycosphingolipids and overexpression of the genes involved in their lysosome metabolism [71]. In this regard, several studies have shown Cer’s involvement in autophagy. Cer indeed induces autophagy through different mechanisms, including the inhibition of Akt and mammalian target of rapamycin (mTOR) and the activation of Beclin-1 (BECN1) [72,73,74]. Moreover, it was recently observed that genetic ablation of *ASAH1* impairs mitochondrial function and autophagy [75]. It was also observed that blocking Cer catalyzed by, for example, the inhibition of SMS with tricyclodecan-9-yl-xanthogenate (D609), promotes an elevation of intracellular Cer, leading to autophagy in hippocampal neurons [76]. Cer was also observed to induce apoptosis by mitophagy. Specifically, CerS1, C_18_-Cer, and pyridinium-C_18_-Cer induce lipidation of the microtubule-associated protein 1 light chain 3 β (LC3B-II, a key protein in autophagosome biogenesis). This molecule binds to Cer in the mitochondrial membrane after mitochondrial fission and directs the autophagy-lysosomes pathway to generate lethal mitophagy [77]. Interestingly, this event can be reverted by the interaction of S1P with S1PR3 [77].

Mitochondrial dysfunction and oxidative stress are closely linked in neurodegenerative processes [78]. In this regard, long-chain and very long-chain Cer can cause mitochondrial dysfunction and oxidative stress [79]. It should be noted that the production of ROS, which is highly implicated in neurodegeneration, is a consequence of mitochondrial dysfunction, leading to cell deterioration and collapse. In neuroblastoma cell line SH-SY5Y, the exposure to Cer induced a dose-dependent increase in free radicals [80]. Similarly, in primary rat hippocampal cell culture, Cer treatment induced a dose-dependent increase in ROS levels [81]. The mitochondrial voltage-dependent anion channel 1 (VDAC1) has been linked to AD pathogenesis [82]. Interestingly, it has recently been shown that the association between Cer and tubulin blocks VDAC1 in astrocytes reduces ATP release and motility [83]. Furthermore, the increase in Cer in dopaminergic neurons (the most affected neuronal type in PD) promotes a loss of mitochondrial membrane potential (MMP) and activation of caspase-3, leading to neuronal apoptosis [69].

In addition to the above considerations, Cer participates in neurodevelopment through different mechanisms. In particular, Cer is involved in the establishment of cell polarity in the ectoderm, regulation of the motility of neuroblasts, and apoptosis processes necessary for the correct development of the CNS [84,85,86]. Significantly, brain ciliogenesis is regulated by nSMase2 activity and Cer by interaction with atypical protein kinase C (aPKC) and GSK3-β [87,88].

S1P is another SL that plays an important role in neurodegenerative diseases. S1P is involved in neuronal survival, modulation of synaptic transmission, and neuroinflammation. In contrast to Cer, S1P promotes cell and neuronal survival, acting as a second messenger of different signaling pathways, such as by activation of ERK or PI3K/Akt pathways, or by inhibition of JNK, caspase-3, or BAX [89,90,91]. However, prolonged exposure to S1P induce apoptosis in hippocampal neurons [92]. Regarding synaptic transmission, S1P has an important role in the CA3 region of the hippocampus. S1P increases the rate of α-amino-3-hydroxy-5-methyl-4-isoxazolepropionic acid receptor (AMPAR) mediated miniature excitatory postsynaptic currents and participates in long-term potentiation (LTP), a fundamental mechanism involved in learning and memory [93]. In line with these data, S1P triggers glutamate secretion in hippocampal neurons and potentiates depolarization-induction secretion of this neurotransmitter [93]. In addition, S1P regulates the localization of synapsin I (a phosphoprotein that promotes the availability of synaptic vesicles) in the presynaptic compartment [94]. Its dephosphorylated analog, Sph, also participates in the tracking of important proteins at the synaptic level by facilitating the assembly of the soluble NSF attachment protein (SNAP) receptor (SNARE) complex and the release of synaptic vesicles [95]. Regarding neuroinflammation, S1P accumulation can activate microglia and, therefore, an inflammatory response, in addition to autophagy alterations [96]. Recently, several reports have shown that the administration of lipopolysaccharide (LPS) contributes to neuroinflammation along with damage to the blood–brain barrier (BBB) [97,98]. In fact, LPS induces S1P production through the SphK1 activation [99]. Moreover, increased S1P levels induces the synthesis of proinflammatory cytokines, such as TNF-α and IL-1β, and the activation of iNOS in microglia [99]. Interestingly, a recent work demonstrated that LPS promotes an elevation of Cer levels by SPT activity [100]. Additionally, it has been observed an astrocytic increased expression of S1PR3 and activation of SphK1 are associated with inflammatory events [101]. Consequently, S1P binds to S1PR3 in an autocrine manner, and in turn induces the expression of different pro-inflammatory mediators, such as cyclooxygenase 2 (COX-2) and IL-6 [101]. Furthermore, S1P also induces astrocyte proliferation and astrogliosis [102]. Interestingly, treatment of glial cultures (astrocytes and microglia) with Fingolimod (also known as FTY720) attenuates the secretion of pro-inflammatory mediators. Thus, S1PR signaling is involved in regulating cytokine and chemokine release in glial cells [103]. Worthy of note is the fact that phosphorylated Fingolimod (pFTY720) is one of the approved treatments to reduce inflammation in MS [103]. S1P, like Cer, participates in autophagy, with different functions upon the cell type. For example, upregulation of SphK1 in neurons induces neuroprotective autophagy, but not in astrocytes [104]. On the other hand, in astrocytes but not in neurons, amino acid starvation induces SphK1-associated autophagy. Interestingly, SphK1 promotes autophagic degradation of huntingtin in astrocytes [104]. In addition, a lack of S1P lyase was associated with an accumulation of Aβ and α-syn, and hyperphosphorylation of tau, leading to neuronal dysfunction and subsequent autophagy [105,106,107].

Overall, different pieces of evidence support the theory that these alterations in SL metabolism are a putative cause or consequence of the onset and progression of different neurodegenerative diseases (Table 1). In fact, the involvement of SLs in these diseases has been extensively studied.

### 3.1. Parkinson’s Disease

In PD, alterations in the levels of Cer, SM, and GM1, among other SLs, have been described in post mortem samples of brain, blood, and cerebrospinal fluid (CSF) [44]. For example, there is a reduction in long-chain Cer (C_22_-Cer, C_23_-Cer, and C_24:1_-Cer) and an increase in CerS1 and CerS4 in the anterior cingulate cortex [108]. Meanwhile, in plasma samples, an increase in different species of Cer (C_16_-Cer, C_18_-Cer, C_20_-Cer, C_22_-Cer, and C_24:1_-Cer) was observed; in this sense, higher levels are associated with worse cognitive functioning [109]. The most common cause of genetic PD is the mutation in the *GBA* gene, which encodes the β-GCase enzyme. As a consequence of this mutation there is an accumulation of GlcCer [110]. Elevated levels of GlcCer increase the aggregation of α-syn and, interestingly, α-syn can reduce the activity of β-GCase, thus establishing a pathological loop [111].

### 3.2. Dementia with Lewy Bodies

In dementia with Lewy bodies (DLB) patients, elevated concentrations of C_16_-Cer, C_20_-Cer, C_18:1_-Cer, and C_24:1_-Cer have been detected in plasma samples compared to controls [112].

### 3.3. Multiple Sclerosis

Regarding the relationship between MS and SLs, elevated levels of C_16_-Cer, C_18_-Cer, C_24_-Cer, and monohexosyl C_16_-Cer were detected in CSF of MS patients, as well as an increase in aSMase activity [113,114]. Interestingly, leukocyte infiltration is a critical step in the pathophysiology of MS. In this regard, CerS2 and CerS6 regulate the levels of chemoattractants, such as C-X-C Motif Chemokine Receptor 2 (CXCR2), that promotes neutrophil migration to the CNS. CerS2 also regulates granulocyte stimulating factor (G-CSF)-induced CXCR2 expression, while CerS6/C_16_-Cer inhibits CXCR2 expression [115,116]. Additionally, in animal models of MS deletion of CerS2 in blood cells resulted in a delay in the onset of symptoms due to reduced infiltration of immune cells into the CNS [117], whereas deletion of CerS6 resulted in an increment in the infiltration [116]. Therefore, targeting CerS2 and CerS6 is a promising target for the treatment of MS.

### 3.4. Amyotrophic Lateral Sclerosis

In the spinal cord of amyotrophic lateral sclerosis (ALS) animal models and patients, an increase in C_16_-Cer, C_24_-Cer, and C_16_-SM levels in the lumbar region has been described. Notably, the accumulation of C_16_-Cer occurred before the onset of symptoms [118]. As for the cervical region, an increase in the total content of Cer, specifically C_18_-Cer, C_24_-Cer, C_24:1_-Cer, and ganglioside 3 (GM3) has been detected [119]. Interestingly, treatment with GM3 slowed the onset of paralysis and increased survival in ASL animal models [119]. Specifically, in FUS transgenic mice, a model of ASL, a reduction in the S1P/Sph-Spha ratio have been detected in the spinal cord, inducing apoptosis in spinal cord cells [120]. In addition, there is a large increase in the expression of S1P lyase in motor neurons decreasing the levels of S1P, which has anti-apoptotic properties [120].

### 3.5. Huntington’s Disease

In post mortem brain analyses of Huntington’s disease (HD) patients, an evident increase in S1P lyase levels in the striatum and cortex was described, along with a decrease in SphK1 in the striatum [121]. Furthermore, the dysregulation in S1P metabolism is an early event observed in both human samples and animal/cellular models [121]. Moreover, downregulation of SPT and CerS, in addition to decreased levels of dihydroSph (dhSph), dhS1P, and dhC_18_-Cer, has been observed in HD mouse models [122]. Notably, treatment with Fingolimod in animal models resulted in improved motor function, reduced brain atrophy, and increased survival [123]. Remarkably, Fingolimod treatment increased the phosphorylation of huntingtin, and consequently reduced its aggregation, resulting in neuronal activity and connectivity improvement. This benefit may also be associated with brain-derived neurotrophic factor (BDNF) levels, as Fingolimod increases BDNF in the CNS [123].

### 3.6. Lysosomal Storage Diseases

Lysosomal storage diseases (LSD) are caused by the metabolic disruption of several molecules that leads to the accumulation of macromolecules in lysosomes. LSD includes different diseases that affect SL metabolism, such as Niemann-Pick’s disease, Gaucher’s disease, Farber’s disease, Krabbe’s disease, and Fabry’s disease. Given that part of the SLs is degraded in lysosomes, one of the key hallmarks of these diseases is neurodegeneration caused by SL metabolism alterations [31,124,125]. Niemann-Pick’s is a genetic disease caused by mutations in the *SMPD1* gene (generating a deficiency of aSMase leading to a progressive accumulation of SM in the organs, including the brain); or in the NPC intracellular cholesterol transporter (*NPC1* or *NPC2)* genes (leading to alterations in cellular cholesterol trafficking) [126]. Gaucher’s disease is due to mutations in the *GBA* gene, resulting in a deficit of the lysosomal enzyme β-GCase and accumulation of GlcCer. This disease is associated with an increased risk of PD and dementia, as *GBA* deficiency increases α-syn aggregates [127,128,129]. Faber’s disease is caused by mutations in the *ASAH1* gene, resulting in a decrease in ASAH1 activity, and an accumulation of Cer [130]. Interestingly, C_26_-Cer was proposed as a diagnostic biomarker of Faber’s disease [131]. Krabbe’s disease is a genetic disease caused by mutations in the *GALC* gene. This results in defects in a defective lysosomal galactosylceramidase (GALC), an essential enzyme for the catabolism of galactocerebroside in order to produce Cer, which is the main lipidic component of the myelin sheath [132].

As already stated, phosphorylated Fingolimod (pFTY720) has been approved for the treatment of MS, a complex disease involving both autoinflammatory and neurodegenerative mechanisms MS [103]. Despite substantial evidence involving SL in other neurodegenerative diseases, no other related treatments have been described. Indeed, the development of new specific drugs that may modify SL metabolism is an attractive therapeutic horizon.

**Table 1 ijms-23-08082-t001:** Summary table of the implication of sphingolipids in neurodegenerative disease. The upward arrows refer to an increase in levels compared to the controls, while the downward arrows refer to a reduction. Acid ceramidase (ASAH1), acid sphingomyelinase (aSMase), ceramide (Cer), ceramide synthase (CerS), galactosylceramidase (GALC), ganglioside 3 (GM3), glucosyl-ceramide (GlcCer), sphingomyelin (SM), sphingosine 1-phosphate (S1P), and sphingosine kinase (SphK).

Disease	Sphingolipid Species	Levels	Source	Implication in the Disease	Ref.
Parkinson’s disease	C_16_-Cer	↑	Plasma	Higher levels are associated with worse cognition	[109]
	C_18_-Cer	↑			
	C_20_-Cer	↑			
	C_22_-Cer	↑			
		↓	Anterior cingulate cortex		[108]
	C_23_-Cer	↓			
	C_24:1_-Cer	↓			
		↑	Plasma	Higher levels are associated with worse cognition	[109]
	CerS1	↑	Anterior cingulate cortex		[108]
	CerS4	↑			
Dementia with Lewy bodies	C_16_-Cer	↑	Plasma		[112]
	C_18:1_-Cer	↑			
	C_20_-Cer	↑			
	C_24:1_-Cer	↑			
Multiple sclerosis	C_16_-Cer	↑	CSF		[113]
	C_18_-Cer	↑			
	C_24_-Cer	↑			
	Monohexosyl C_16_-Cer	↑			
	aSMase activity	↑			[114]
Amyotrophic Lateral Sclerosis	C_16_-Cer	↑	Spinal cord lumbar region	Accumulation before the onset of the symptoms	[118]
	C_18_-Cer	↑	Spinal cord cervical region		[119]
	C_24_-Cer	↑	Spinal cord lumbar region		[118]
		↑	Spinal cord cervical region		[119]
	C_24:1_-Cer	↑			
	C_16_-SM	↑	Spinal cord lumbar region		[118]
	GM3	↑	Spinal cord cervical region		[119]
Huntington’s disease	S1P lyase	↑	Striatum and cortex		[121]
	SphK1	↓	Striatum		
Niemann-Pick’s disease type A and B	SM	↑	Systemic organs and brain	Disease caused by a mutation in *SMPD1* gene generating a deficiency of aSMase and the accumulation of SM	[126]
Gaucher’s disease	GlcCer	↑	Macrophage lysosomes	Disease caused by a mutation in *GBA* gene generating a deficiency of β-GCase and the accumulation of GlcCer	[127]
Faber’s disease	Cer	↑	Systemic organs and brain	Disease caused by a mutation in *ASAH1* gene generating a decreasing of ASAH1 activity and the accumulation of Cer	[130]
	C_26_-Cer	↑	Blood	Proposed as a diagnostic biomarker	[131]
Krabbe’s disease	galactocerebrosides	↑		Disease caused by a mutation in *GALC* gene generating a deficiency of GALC and the accumulation of galactosphingolipids (including galactocerebrosides)	[132]

## 4. Ceramide Metabolism in Alzheimer’s Disease

Different alterations in the SL metabolism have been described in AD. From the studies carried out to date, it can be concluded that Cer and S1P play a pivotal role in several pathophysiological processes of this disease. In this section, we will discuss the genetic changes in SL metabolism and their implication in the pathophysiology of AD.

### 4.1. Gene Expression of Enzymes Involved in Sphingolipid Metabolism in Alzheimer’s Disease

Several analyses in brain samples from AD patients have observed changes in genes related to enzymes involved in the SL metabolism. For example, in de novo pathway, the expression of *serine palmitoyltransferase long chain base subunit 2* (*SPTLC2*) is up regulated in severe AD [25]. In addition, *CERS-1* and *-2* genes were found to be up regulated, while *CERS-6* was observed to be down regulated in mild and severe AD [25]. In this regard, overexpression of *SMPD1*, *SMPD2* (coding for aSMase and nSMase2, respectively), and *GALC* was detected in AD brain samples [27]. In addition, *ASAH1* was shown to be down regulated in mild and severe AD patients [25].

Phosphatidic acid phosphatase type 2B (PPAP2B) enzyme catalyzes the hydrolysis of C1P and S1P to Cer and Sph, respectively. Remarkably, *PPAP2B* gene, as well as *SGPL1*, a gene encoding for S1P lyase 1, are significantly up regulated in the early and severe stages of AD [25].

Genetic alterations may also lead to an abnormal expression of enzymes related to complex SLs, such as cerebrosides and gangliosides. In this regard, it is well known that mutations in *GBA* (which encodes for β-GCase) are related to other diseases with neurological involvement such as Gaucher’s disease and PD [129]. In addition, the expression of UDP-glucose ceramide glucosyltransferase (*UGCG*), an enzyme that catalyzes the first glycosylation in cerebroside biosynthesis, was described to be significantly decreased in all stages of AD [25]. In contrast, *UGT8* gene (encoding the ceramide UDP-Galactosyltransferase, a key enzyme step in the biosynthesis of galactocerebrosides) was found to be up regulated only in severe AD [25]. Regarding gangliosides, the expression of *β-1,4-Galactosyltransferase 6* (*B4GALT6)* progressively decreases as the disease progresses [25].

Considering the above mentioned, SL metabolism enzymes alterations observed in different stages of AD lead to an increase in Cer levels (proapoptotic) and a decrease in S1P and C1P (anti-apoptotic) [54,91], which may determine or promote cell fate in the pathophysiology of AD.

Interestingly, a recent in silico study identified three genes involved or related to SL metabolism and distribution (*SPHK1*, *CAV1,* and *SELPLG*) with the potential to modify the expression levels of some genes involved in the disease phenotype [24]. *SPHK1* gene encodes the previously described SphK1. *CAV1* encodes for caveolin-1, a protein that interacts with cholesterol and SM; it is involved, among other functions, in the structural composition of lipid rafts. Meanwhile, the *SELPLG* gene (E-selectin receptor) promotes signal transduction through interactions with membrane [24].

### 4.2. Role of Ceramide in the Pathophysiology of Alzheimer’s Disease

Several works have addressed the implication of Cer in AD progression, as it is involved in the formation of Aβ and consequently in the neurotoxicity that this peptide exerts throughout the brain. Remarkably, Aβ itself can induce the synthesis of Cer, establishing a pathogenic cycle that induces the aggregation of amyloid plaques and the progression of AD. Other pathophysiological events in AD are tau hyperphosphorylation, ROS production, mitochondrial dysfunction, lipid peroxidation, and neuroinflammation, where Cer was also found to be involved.

Amyloid plaques, due to Aβ aggregation, is one of the hallmarks of AD. Aβ originates by cleavage of its precursor, the amyloid precursor protein (APP), a transmembrane protein that is expressed in many tissues. APP is especially found in neuronal synapses, where it acts as a regulator of synaptic formation and repair, among other functions. APP can be processed through two different pathways depending on the action of different enzymes: the non-amyloidogenic pathway, and the amyloidogenic pathway. In the non-amyloidogenic pathway, APP is initially cleaved by α-secretase releasing the N-terminal sAPPα fragment from the membrane and generating the membrane-tethered α-C terminal fragment (CTFα). Then, CTFα is cleaved by γ-secretase resulting in the extracellular peptide P3 and APP intracellular domain (AICD), soluble products not related to amyloid plaque formation. However, when APP is initially processed by β-secretase (BACE-1), Aβ formation is induced, namely, BACE-1 generates CTFβ and sAPPβ. CTFβ is then processed by γ-secretase into extracellular space producing Aβ and AICD. In this scenario, the final Aβ product is not soluble and eventually aggregates to form the amyloid plaques (Figure 2A) [133].

Concerning Cer and Aβ synthesis, Cer was shown to stabilize and increase the half-life of BACE-1, promoting the formation of Aβ [134]. This effect was observed using short Cer analogs (C_6_-Cer) and nSMase treatments [134]. Significantly, CerS inhibition by fumosin B1 decreases Aβ production [134]. Moreover, different Cer analogs act as modulators of γ-secretase activity and also increase Aβ levels [135]. The p75 neurotrophin receptor (p75NTR) was described to enhance Aβ formation by activating APP processing by BACE-1. Interestingly, p75NTR acts activating Cer as a second messenger of this pathway [136]. Tyrosine kinase receptor A (TrkA) is another neurotrophin receptor, which, conversely, reduces APP processing by BACE-1 [136]. During normal aging, there is a switch from TrkA to p75NTR receptor system that increases Aβ levels [136]. Interestingly, nSMase inhibition by manumycin A can induce a change in this switch [136].

Other SLs have been also observed to induce Aβ formation, such as S1P, which increases Aβ production by regulating BACE-1 activity at the neuronal level [137]. On one hand, both pharmacological inhibition by sphingosine kinase inhibitor (SKI) II, SKI V, and *N,N*-dimethylsphingosine and RNA interference knockdown of SphK reduced Aβ synthesis [137]. On the other hand, S1P lyase plays a fundamental role in the regulation of APP metabolism. Inhibition of S1P lyase and the consequent rise in S1P levels increases the production of APP-CTFs α and β in lysosomal compartments, whereas inhibition of SphK decreases their production [106]. Moreover, in S1P lyase knock-out cells there is a decrease in Aβ/APP ratio concentration and lower γ-secretase activity [106]. As discussed above, Cer can induce Aβ formation [134,135,136,138]. However, Aβ was also defined to induce Cer formation by activating SMases [139,140,141,142]. Thus, a vicious circle in which Cer and Aβ levels are increased is established.

In addition, Cer plays an important role in establishing the neurotoxicity of Aβ by induction of apoptosis in neurons, astrocytes, and oligodendrocytes. In neuronal cultures, Aβ activates aSMase and nSMase, with subsequent Cer formation (Figure 2B) [139,140]. This process involves NADPH oxidase-superoxide-hydrogen peroxide and the cytosolic calcium-dependent phospholipase A2 (cPLA2)-arachidonic acid (AA) pathway, leading to neuronal death via apoptosis [139,140]. Moreover, Aβ was shown to down regulate the activity of SphK1, increasing the Cer/S1P ratio and, consequently, neuronal death in SH-SY5Y neuroblastoma cells culture (Figure 2B) [143]. Aβ can also induce neuronal apoptosis through the activation of astrocytes, both in vivo and in vitro. Aβ_1–42_ activates astrocytes, which secrete different pro-inflammatory molecules, such as NO, that induce nSMase activation in neurons with an increase in Cer levels leading to apoptosis [141]. In this sense, inactivation of nSMase by antisense oligonucleotides, or GW4869 (a specific inhibitor of nSMase) protects neurons from Aβ toxicity and prevents its activation by pro-inflammatory cytokines in vitro and in vivo [141]. Interestingly, S1P also has a neuroprotective effect by inhibiting aSMase activation by Aβ and apoptosis [140].

Moreover, Cer also induces Aβ-mediated apoptosis in astrocytes and oligodendrocytes through activation of nSMase [142,144]. It has been reported that Aβ_42_ induces an elevation of Cer levels in astrocytes, by activation of nSMase2, resulting in mitochondrial fragmentation [83]. Voltage-dependent anion channel-1 (VDAC1) has been linked to AD pathogenesis and was found at high levels in post mortem brains of AD patients [82]. It has recently been observed that the interaction of Cer with tubulin may lead to an interaction with VDAC1 in the outer mitochondrial membranes, producing a depletion in ATP levels (Figure 2B). In addition, fumosin B1 treatment or nSMase2-deficient astrocytes prevent mitochondrial alterations [83]. Recently, a new mechanism has been proposed to reduce Aβ neurotoxicity through Cer transfer proteins (CERT). In this regard, induction of a long isoform of CERT expression in the brain of the 5xFAD mice (an AD model) decreased C_18:1_ and C_16_-Cer, with a subsequent decrease in Aβ formation and the microglial pro-inflammatory phenotype [145].

Several studies have described the link between lipid peroxidation and the development of AD [146]. Remarkably, Aβ-induced neuronal accumulation of Cer has also been associated with lipid peroxidation. Specifically, Aβ_1–42_ induces accumulation of C_18_-Cer and C_24_-Cer and decreased C_24_-SM in hippocampal neurons in vitro, together with increased 4-Hydroxynonenal (HNE), a product of lipid peroxidation [147]. Both Aβ and TNFα in vivo treatments have been described to induce an increase in lipid peroxidation and activation of nSMase, with the subsequent elevation of Cer (Figure 2C) [148]. In this sense, inhibition of SPT by myriocin and pre-treatment with a well-known antioxidant, such as α-tocopherol, significantly reduced Cer levels, lipid peroxidation products, and Aβ-induced neuronal death in vitro [147].

Another important event that occurs in AD, in addition to the Aβ and tau accumulation, is neuroinflammation. A recent genetic study of more than 100,000 AD patients found 75 high-risk genes for the disease, identifying 42 more than those already established. It is noteworthy that more than 20 of those genes affect microglia [149]. In this regard, several works have addressed the implication of Cer and S1P in pro-inflammatory pathways.

As explained, Aβ induces the activation of nSMase in astrocytes (Figure 2D). This activation with an elevation in Cer levels promotes the activation of NF-κB and iNOS, leading to an increase in NO and glial fibrillary acidic protein (GFAP), a marker of glial activation [141]. Consequently, many pro-inflammatory molecules such as TNFα, IL-6, IL-1β, and NO are released. However, as previously described, the inhibition of nSMase with GW4869 and antisense oligos may provide rescue from this pro-inflammatory phenotype [141]. How Aβ stimulates iNOS activity through TNF-α has been previously described. Subsequently, it was observed that the overexpression of truncated nSMase considerably decreased iNOS activity, suggesting its involvement in the pro-inflammatory phenotype of oligodendrocytes [150].

Small molecules capable of specifically inhibiting certain enzymes of SL metabolism have recently been developed and characterized. ARN14494 is a molecule that specifically inhibits SPT. In primary mouse astrocytes, treatment with ARN14494 showed its ability to prevent the synthesis of proinflammatory molecules, oxidative stress related enzymes, such as iNOS or cyclooxygenase-2 (COX2), and to protect neurons from Aβ toxicity [151].

S1P was also shown to be involved in microglia-mediated neuroinflammation. For example, S1P accumulation activates microglia, and it is linked to an impairment of autophagy and inflammation [96]. Moreover, S1PR2 and 3 have been involved in microglial activation after cerebral ischemia through activation of the ERK1/2 pathway [152,153]. The triggering receptor expressed on myeloid cells 2 (TREM2) have been revealed as a key player in microglial activation and AD, and even as a biomarker for the disease [154]. Interestingly, a recent work has demonstrated that S1P can act as endogenous ligand of TREM2 to activate microglial phagocytosis [155]. Additionally, SL transporter 2 (Spn2) promotes the release of pro-inflammatory cytokines in microglia induced by S1P and NF-κB pathway [156]. In addition, Spns2-knockout mice significantly reduce Aβ_1–42_-induced microglia activation in vivo and in vitro and improve cognition in vivo [156]. Furthermore, high expression of metalloproteinases has been described in several neurological disorders, including AD [157]. Following this line, Ordoñez et al. demonstrated that C1P stimulates macrophages migration by a mechanism that involved metalloproteinases 2 and 9 [158]. Moreover, Cer has been observed to stimulate metalloproteinase-9 release in airway epithelium cells by activation of Janus kinase 2 (JAK2)/Signal transducer and activator of transcription 3 (STAT3) pathway [159], a well-known involved pathway in AD [160].

As discussed above, tau hyperphosphorylation is another hallmark of AD and an activator of microglia [161]. Interestingly, a recent study demonstrated that tau-activated microglia and neurotoxicity were blocked by inhibition of nSMase2 with a GW4869 inhibitor [162]. Furthermore, another recent work has shown that inhibition of SPT with myriocin increased tau hyperphosphorylation in yeast [163]. Moreover, in S1P-lyase-deficient mice an increase in S1P levels accompanied by tau hyperphosphorylation in the hippocampus and cortex has been observed [105].

Exosomes are extracellular vesicles that are characterized by transport molecules such as RNA or proteins for intercellular communication. In recent years, their involvement in the development of AD has been evidenced [164,165]. It is well known that the inhibition of nSMase by GW4869 blocks the exosome release [166]. On the other hand, it has been demonstrated that the administration of plant-derived GlcCer stimulates the release of extracellular vesicles [167]. This leads to increased expression of neuronal markers and alleviation of pathology in APP^Swe/Ind^ transgenic mice. Interestingly, these plant-derived sphingolipids have recently been shown to be able to cross the BBB [168].

Interestingly, astrocytes that release Cer-enriched exosomes have been shown to accelerate Aβ aggregation and decrease their glial clearance [138]. Furthermore, genetic deletion of nSMase2 entailed decreased levels of Cer and exosomes, glial activation, Aβ42 and amyloid plaques, tau phosphorylation, and improved cognition in an AD 5XFAD animal model [138]. Cer-enriched exosomes derived from astrocytes are not only involved in amyloid plaque formation but are also capable of inducing apoptosis and ROS [142].

Moreover, Aβ-associated-Cer-enriched astrocyte-derived exosomes induce Aβ binding to the mitochondrial VDAC1 in neurons. This results in an altered mitochondrial metabolism, activation of caspases, neurite fragmentation, and neuronal death (Figure 2B) [142,169].

## 5. Sphingolipidomics in Alzheimer’s Disease

In AD, several studies have reported alterations in the SL profile, observed in the brain tissue, CSF, and blood (Table 2). These changes have been observed in the different stages of AD, including early stages such as mild cognitive impairment (MCI). In fact, as the disease progresses, different changes in SL levels, as well as in enzymes involved in their metabolism have been detected.

### 5.1. Brain Tissue

Interestingly, during physiological aging, both Cer and cholesterol accumulate in neuronal cells, associated with oxidative stress [147]. In this regard, concentrations of C_24_-Cer, and complex SL C_24_-GalactosylCer (GalCer) were found to be increased in brain cortex of aging C57BL/6 mice [147]. A trend of higher total Cer, SM, and Sph content was also reported in human hippocampus [170]. In addition, in the hippocampus, the S1P/Sph ratio correlates negatively with age [170]. This is interesting since aging is a risk factor for AD, which also correlates with increased Cer levels.

A significant increase in total Cer levels and reduction in SM compared with healthy controls was described in brain samples from AD patients [171]. In particular, higher levels of C_16_, C_18_, C_20_, and C_24_-Cer have been detected [27]. These changes in brain tissue from AD subjects have been observed in different areas. In the middle frontal gyrus, there is an increase in the concentration of C_24_-Cer and GalCer and a reduction in SM, however, these alterations were not observed in non-vulnerable brain regions of AD patients, such as the cerebellum [147]. Regarding lipid peroxidation, elevated HNE levels were also detected in the middle frontal gyrus, associated with lipid membrane oxidative stress [147]. At the prefrontal cortex, an increased concentration of Cer (mostly C_18:1/18:0_-Cer), GlcCer, and GalCer was also detected [172]. Regarding the white matter, one study detected elevated levels of Cer with a peak concentration at the very early AD stage, mainly at the temporal region [173]. Interestingly, the very long-chain Cer C_24:1_-Cer showed the greatest increase, which is consistent with previously mentioned data [147,173]. In addition, no changes in GalCer or SM levels were detected in white matter from the superior temporal cortex, except in severe AD where there were slight alterations [173]. Regarding SM, at the prefrontal cortex, reduced levels of medium-length chain SM (C_18:0/20:0_ and C_18:1/20:0_-SM) and increased levels of long-chain SM (C_18:1/22:1_ and C_18:0/26:1_-SM) were detected; whereas at the entorhinal cortex, elevated levels of short-chain SM (C_18:0/18:0_, C_18:1/16:1_, C_18:1/18:0_-SM) were observed [172]. As described, SLs are the main component of cell membranes. When analyzing the composition of cell membranes extracted from brains at different disease stages of AD, a higher composition of long-chain Cer (such as C_18_ and C_24_-Cer) was detected when compared with age-matched controls [147]. Interestingly, the highest values corresponded to the greatest severity of AD [147]. In this sense, this overall increase in Cer and decrease in SM levels may be explained by the increased expression of enzymes, such as aSMase in AD brains [171]. A significant correlation was observed between brain levels of aSMase, Aβ, and hyperphosphorylated tau, consistent with Cer synthesis induced by Aβ through SMase activity in vitro [171]. Interestingly, during normal aging, the activity of nSMase increases in areas associated with behavior, such as the striatum, the prefrontal cortex, and especially in the hippocampus, which is considered a main region affected in AD [174]. In addition, the increased activity of this enzyme correlated with increased expression of pro-inflammatory markers such as IL-1β [174]. Continuing with the relationship between Aβ and Cer, in the frontal cortex of AD patients, Cer-immunoreactive astroglia has been detected in amyloid plaques [175]. Moreover, amyloid plaques were described to be enriched in C_18:1/18:0_ and C_18:1/20:0_-Cer, probably due to the presence of nSMase and aSMase in the corona of amyloid plaques [176]. Additionally, in reactive astrocytes and microglia associated with cerebral amyloid angiopathy (Aβ accumulation around capillaries), there is an increased expression of Cer and SMases, besides S1PR1 and S1PR3 [177].

Furthermore, increased levels of both Sph and ASAH1 were also detected in AD brains compared to healthy controls [171]. Moreover, the concentration of S1P, was found decreased in the frontotemporal region [171]. Interestingly, this study has shown a negative correlation between Aβ, hyperphosphorylated tau, and S1P levels [171]. In fact, in the entorhinal and frontal cortex of AD patients, a lower expression of SphK1 and S1PR_1_ was described, along with a higher expression of S1P lyase [178]. Furthermore, changes in SphK1 and S1P lyase expression were correlated with amyloid deposits in the entorhinal cortex [178].

It is well known that SphK2 is located in both cytosol and nuclei, while cytosolic SphK2 is related to proliferation and cancer [179], and nuclear SphK2 is involved in gene expression and telomere integrity [180]. Recently, Dominguez and co-workers demonstrated that there is a decrease in cytosolic SphK2 and an increase in nuclear SphK2 in the hippocampus and frontal cortex [181]. These authors hypothesized that this would lead to a detriment of cytosolic S1P and a subsequent decrease in their pro-survival capacities [181]. The presence of truncated SphK2, which is associated with apoptosis at both nuclear and cytosolic levels, was also increased in AD brains [181,182].

### 5.2. Cerebrospinal Fluid (CSF)

In CSF, elevated levels of Cer have been detected in all stages of AD. Paradoxically, concentrations were found to be higher in moderate AD compared to mild and severe stages of the disease [175]. Curiously, Cer levels in CSF were higher in AD patients than in other neurological diseases [175]. Indeed, C_18_-Cer levels in CSF were significantly associated with the presence of classical AD markers, such as Aβ_42_ and total tau, as well as inflammatory markers, such as S100 calcium-binding protein B (S100B) in the earlier stages of dementia [183]. In a fraction analysis of CSF, Cer was detected in the nanoparticle fraction (indicative of vesicular membrane metabolism) and the supernatant fraction (representative of interstitial metabolism) in patients with probable AD, MCI, and healthy controls [184]. The SM/Cer ratio was higher in the supernatant fraction in probable AD compared to healthy controls, whereas Cer levels were lower in the nanoparticle fraction [184]. Conversely, aSMase, but not nSMase, activity was decreased in CSF from AD subjects compared to controls and MCI patients [184]. Contrarily, elevated SM levels were detected in prodromal AD compared to healthy controls, whereas in mild AD a decrease in C_18:1/14:0_ and C_18:1/16:0_-SM concentration was observed [185]. Moreover, a significant decrease in C_24:1_-SM and an increase in C_24:0_-Cer levels was detected in AD compared to healthy subjects and patients with idiopathic normal pressure hydrocephalus (iNPH, a subcortical dementia and gait impairment) [186]. In addition, decreased S1P and increased C_24:0_-GlcCer levels were detected in AD compared to iNPH [186]. Recently, another study concluded that no SMs were associated with diagnostic biomarkers of AD or MCI in CSF [187]. However, some SMs (C_18:1/18:0_, C_18:2/16:0_, C_18:1/16:1_, C_18:1/18:1_, C_18:2/18:0_, C_18:1/14:0_, C_16:1/16:0_, and C_18:1/16:0_-SM) did associate with biomarkers of neurodegeneration such as p-tau181, α-syn, neurogranin, and neurofilament light chain (NfL), as well as neuroinflammatory biomarkers, such as chitinase-3-like-protein 1 (YKL40) or soluble TREM2 (sTREM2), indicating their potential to be biomarkers for some of these pathological situations [187]. As for S1P, its concentration in CSF increases progressively from healthy controls to MCI patients, while its levels decrease in AD subjects, this being consistent with data obtained in post mortem brain samples of AD patients [188].

### 5.3. Blood

Interestingly, elevated levels of plasma C_16_, C_21_, C_18:1_, C_20_, C_21,_ and C_24:1_-Cer as well as C_18:1_ and C_24:1_-hexosylceramide (HexCer) have been detected in AD patients compared with controls [112,189]. Moreover, very long-chain C_22_- and C_24_-Cer in plasma were lower in MCI, compared with patients with AD or controls; in addition, higher levels in MCI patients were predictive of cognitive decline [190]. In this sense, elevated serum levels of C_18:1/16:0_- and C_18:1/24:0_-Cer, and LacCer were found to be associated with a 7 to 10-fold increase in AD risk in elderly women without dementia [191]. However, in another study, no relationship between different plasma Cer species and the risk of AD in women was found, whereas elevated levels of C_16:0_-Cer, C_18:0_, C_18:1_, C_20:1,_ and C_22:1_-SM increase the risk of AD in men [192]. Interestingly, elevated serum concentrations of Spha 1-phosphate (Spha-1P) were also described to predict the conversion of MCI to AD [193].

Regarding SM, a reduction in C_22:1_- and C_24:1_-SM plasma concentration was observed [189]. In this sense, a significant increase in the levels of long- and short-chain Cer was detected in serum samples, along with an increase in short-chain SM [186]. Interestingly, a recent study demonstrated a correlation between Aβ_1–42_ levels in CSF and GlcCer in serum, while total tau in CSF and brain atrophy correlated with Cer and monounsaturated SM concentration in serum [194]. Surprisingly, elevated plasma levels of SM in women were associated with a lower risk of AD [192]. This association was stronger among APOEε4 carriers; conversely, no association was detected between the presence of APOEε4, Cer levels, and increased risk of AD [192].

As described, alterations in both plasma and serum SL levels may predict different neurological alterations. Elevated plasma C_22_- and C_24_-Cer levels predict hippocampal volume loss and cognitive impairment in MCI patients [190]. Increased plasma SM/Cer and dihydro-sphingomyelin (dhSM)/dihydro-ceramide (dhCer) ratios were observed to indicate a slow progression of cognitive decline in AD [195]. Moreover, elevated serum levels of SM and total Cer predicted cognitive impairment up to nine years later since baseline studies, whereas low levels were associated with cross-sectional impairment of delayed recall memory [196]. Interestingly, C_16:0_, C_20:0_-Cer levels predicted impaired immediate recall and psychomotor speed. Thus, Cer serum levels vary according to the onset of memory impairment and could be used as biomarkers of early stages of AD [196].

**Table 2 ijms-23-08082-t002:** Summary table of human sphingolipidomics. The upward arrows refer to an increase in levels compared to the controls, while the downward arrows refer to a reduction. The asterisk indicates non statistically significant data. Ceramide (Cer), hexosyl-ceramide (HexCer), galactosyl-ceramide (GalCer), glucosyl-ceramide (GlcCer), lactosyl-ceramide (LacCer), sphingomyelin (SM), dihydro-sphingomyelin (dhSM), dihydro-ceramide (dhCer), sphingosine (Sph), sphingosine 1-phosphate (S1P) and sphinganine 1-phosphate (Spha-1P), Alzheimer’s disease (AD), clinical dementia rating (CDR), cerebrospinal fluid (CSF), Mini-Mental State Examination (MMSE), idiopathic normal pressure hydrocephalus (iNPH), β-amyloid (Aβ) S100 calcium-binding protein B (S100B), Lewy bodies (LB), dementia with Lewy bodies (DLB), and ε4 allele of the apolipoprotein E gene (*APOEε4*).

Sphingolipid Species	Levels	Source	AD Stage/Condition	Notes	Ref.
**Total Cer**	↑ *	Hippocampus	Cognitively normal elderly	Increased trend with age	[170]
	↑		Cognitively normal elderly men	Significantly correlated with age in males	
	↑	Brain samples	Not specified		[171]
	↑	Prefrontal cortex			[172]
	↑	Temporal, frontal, and parietal white matter	Very early (CDR = 0.5), mild (CDR = 1), moderate (CDR = 2), and severe AD (CDR = 3)	Peak concentration at very early AD in temporal with matter	[173]
	↑	Frontal cortex	Not specified	Detection of Cer-immunoreactive astroglia in amyloid plaques in layer 2 and 3 of the frontal cortex	[175]
	↑	Occipital cortex	Braak stages 4 to 6	Increased expression of Cer in reactive astrocytes and microglia associated with cerebral amyloid angiopathy	[177]
	↑	CSF	Not specified	Higher concentration in moderate AD compared to mild and severe stages	[175]
	↓	CSF nanoparticle fraction	Probable AD		[184]
	↑	CSF supernatant fraction			
	↑	Serum	Not specified	Correlation with total tau levels in CSF and brain atrophy	[194]
	↑		Elderly women without dementia (MMSE score ≥ 24)	Predicted cognitive impairment in asymptomatic individuals	[196]
	↓			Association with cross-sectional impairment of delayed recall memory	
**Short-chain Cer**	↑		Not specified	Compared to healthy controls and iNPH patients	[186]
**Long-chain Cer**	↑				
**C_16_-Cer**	↑	Frontal cortex	Braak stages 1 to 6		[27]
	↑	Hippocampus	Cognitively normal elderly	Associated with age	[170]
	↑	Plasma	Mild and moderate (MMSE ≥ 20) AD		[189]
	↑		AD (Braak stage ≥ 4 and no LB) and high-likelihood DLB (Braak stage ≤ 4)	Compared to cognitively normal controls	[112]
	↑		Elderly men	Increased risk of AD	[192]
	↑	Serum	Elderly women without dementia (MMSE score ≥24)	Prediction of impaired immediate recall and psychomotor speed	[196]
**C_18_-Cer**	↑	Frontal cortex	Braak stages 1 to 6		[27]
	↑	Middle frontal gyrus	Mild (MMSE 23–29), moderate (MMSE 11–20) and severe (MMSE 0–10) AD patients	Analysis of cell membranes. Highest values corresponded to greatest severity of AD	[147]
	↑	CSF	MCI (MMSE 24–30)	Significantly associated with AD (Aβ_42_ and total tau) and inflammatory (S100B) markers	[183]
**C_18:1_-Cer**	↑	Plasma	AD (Braak stage ≥ 4 and no LB) and high-likelihood DLB (Braak stage ≤ 4)	Compared to cognitively normal controls	[112]
**C_18:1/16:0_-Cer**	↑	Serum	Elderly women without dementia (MMSE score ≥ 24)	Significantly associated with a 7 to 10-fold increase in the risk of AD	[191]
**C_18:1/18:0_-Cer**	↑	Prefrontal cortex	Not specified		[172]
	↑	Superior temporal gyrus	Braak stage 6	Detection of Cer-enriches amyloid plaques	[176]
**C_18:1/20:0_-Cer**	↑				
**C_18:1/24:0_-Cer**	↑	Serum	Elderly women without dementia (MMSE score ≥ 24)	Significantly associated with a 7 to 10-fold increase in the risk of AD	[191]
**C_20_-Cer**	↑	Frontal cortex	Braak stages 1 to 6		[27]
	↑	Serum	MMSE score ≥ 24	Prediction of impaired immediate recall and psychomotor speed	[196]
	↑	Plasma	AD (Braak stage ≥ 4 and no LB) and high-likelihood DLB (Braak stage ≤ 4)	Compared to cognitively normal controls	[112]
**C_21_-Cer**	↑		Mild and moderate (MMSE ≥ 20) AD		[189]
**C_22_-Cer**	↑	Hippocampus	Cognitively normal elderly	Associated with age	[170]
	↓	Plasma	MCI (CDR = 0.5)	Compared with AD (CDR = 1) patients and controls	[190]
	↑			Higher level predicted hippocampal volume loss and cognitive impairment	
**C_24_-Cer**	↑	Frontal cortex	Braak stages 1 to 6		[27]
	↑	Middle frontal gyrus	Mild (MMSE 23–29), moderate (MMSE 11–20) and severe (MMSE 0–10) AD patients	Analysis of cell membranes. Highest values corresponded to greatest severity of AD	[147]
	↑ *	CSF	Not specified	Compared to iNPH patients	[186]
	↓	Plasma	MCI (CDR = 0.5)	Compared with AD (CDR = 1) patients and controls	[190]
	↑			Higher level predicted hippocampal volume loss and cognitive impairment	
**C_24:1_-Cer**	↑	Temporal white matter	Very early AD		[173]
	↑	Plasma	AD (Braak stage ≥ 4 and no LB) and high-likelihood DLB (Braak stage ≤ 4)	Compared to cognitively normal controls	[112]
**C_18:1_-HexCer**	↑				
**C_24:1_-HexCer**	↑				
**GalCer**	↑	Prefrontal cortex	Not specified		[172]
**C_24_-GalCer**	↑ *	Middle frontal gyrus	Mild (MMSE 23–29), moderate (MMSE 11–20) and severe (MMSE 0–10) AD patients		[147]
	↑	CSF	Not specified	Compared to iNPH patients	[186]
**GlcCer**	↑	Serum	Not specified	Correlation between Aβ_1–42_ levels in CSF	[194]
**LacCer**	↑		Elderly women without dementia (MMSE score ≥24)	Significantly associated with a 7 to 10-fold increase in the risk of AD	[191]
**Total SM**	↑ *	Hippocampus	Cognitively normal elderly	Increased trend with age. Significantly correlated with age in males	[170]
	↑ *		Cognitively normal elderly men	Significantly correlated with age in males	
	↓	Brain samples	Not specified		[171]
	↓	Middle frontal gyrus	Mild (MMSE 23–29), moderate (MMSE 11–20) and severe (MMSE 0–10) AD patients		[147]
	↑	CSF	Prodromal AD (MMSE 24–29)	Compared to cognitively normal controls	[185]
	↓	CSF nanoparticle and supernatant fraction	Probable AD		[184]
	↑	Plasma	Elderly men	Increased risk of AD	
	↑		Elderly women	Association with lower risk of AD. Greater association among APOEε4 carriers	[192]
	↑	Serum	Elderly women without dementia MMSE score ≥ 24	Predicted cognitive impairment in asymptomatic individuals	[196]
	↓			Association with cross-sectional impairment of delayed recall memory	
**Monounsaturated SM**	↑		Not specified	Correlation with total tau levels in CSF and brain atrophy.	[194]
**Short-chain SM**	↑			Compared to healthy controls and iNPH patients	[186]
**C_18:0_-SM**	↑	Plasma	Elderly men	Increased risk of AD	[192]
**C_18:1_-SM**	↑				
**C_18:0/18:0_-SM**	↑	Entorhinal cortex	Not specified		[172]
**C_18:1/14:0_-SM**	↓	CSF	Mild AD (MMSE 21–23)	Compared to cognitively normal controls	[185]
**C_18:1/16:0_-SM**	↓				
**C_18:1/16:1_-SM**	↑	Entorhinal cortex	Not specified		[172]
**C_18:1/18:0_-SM**	↑				
**C_18:0/20:0_-SM**	↓				
**C_18:1/20:0_-SM**	↓				
**C_18:1/22_:SM**	↑				
**C_18:0/26:1_-SM**	↑				
**C_20:1_-SM**	↑	Plasma	Elderly men	Increased risk of AD	[192]
**C_22:1_-SM**	↓	Plasma	Mild and moderate (MMSE ≥ 20) AD		[189]
	↑		Elderly men	Increased risk of AD	[192]
**C_24:1_-SM**	↓	CSF	Not specified	Compared to healthy controls and iNPH patients	[186]
	↓	Plasma	Mild and moderate (MMSE ≥ 20) AD		[189]
**SM/Cer**	↑	Plasma	Not specified	Correlation with slow progression of cognitive decline	[195]
**dhSM/dhCer**	↑				
**Total Sph**	↑ *	Hippocampus	Cognitively normal elderly	Increased trend with age. Significantly correlated with age in male	[170]
	↑	Brain samples	Not specified		
**S1P**	↓			Negative correlation between Aβ, hyperphosphorylated tau, and S1P levels	
	↓	CSF		Compared to iNPH patients	[186]
	↓		Mild AD		[188]
	↑		MCI	Increased progressively concentration from healthy controls to MCI patients	
**S1P/Sph**	↑	Hippocampus	Cognitively normal elderly women	Inversely correlated with age	[170]
**Spha-1P**	↑	Serum	MCI	Prediction of the conversion of MCI to AD	[193]

## 6. Concluding Remarks

Nowadays, the origin of AD is still unknown in most cases, and its diagnosis is delayed. Different studies, both molecular and clinical, have evidenced the involvement of SLs in the development of AD. In addition, in recent years, with the improvement of the detection of SLs by mass spectrometry, a correlation has been observed between certain species of SLs, both in brain tissue and in different fluids, and the particular stage of the disease. As noted throughout this work, significant variations of the different species of sphingolipids have been detected in different neurodegenerative diseases. Since biomarkers for early detection of these diseases have not yet been developed, the study of the ability of SLs to become these early biomarkers holds promise for the future. It has also been widely demonstrated that SLs are involved in tau hyperphosphorylation, Aβ aggregation, glia activation, and BBB dysfunction. However, due to the complexity of SL metabolism, in-depth preclinical and clinical studies are still required to elucidate their involvement in neurodegenerative processes. This knowledge may also result in the development of new therapeutical strategies, a clear example of this potential being the approval of Fingolimod for the treatment of MS.

## Figures and Tables

**Figure 1 ijms-23-08082-f001:**
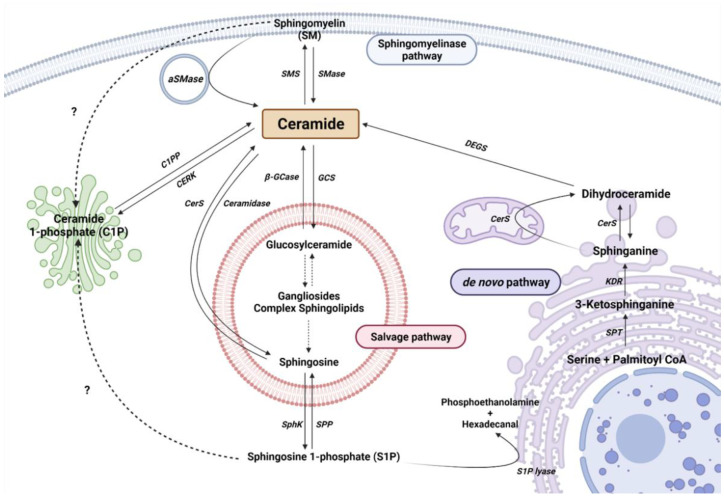
Sphingolipid metabolism. Single reactions are represented by solid arrows, while multiple-step reactions are shown as dashed arrows. Interrogation marks with dashed arrows point to possible mechanisms not yet described. Serine palmitoyltransferase (SPT), 3-keto-dihydrosphingosine reductase (KDR), ceramide Synthase (CerS) and dihydroceramide desaturase (DEGS), sphingomyelinase (SMase), acid sphingomyelinase (aSMase), sphingomyelin synthase (SMS), acid β-glucosidase (β-GCase), glucosylceramide synthase (GCS), ceramide synthase (CerS), sphingosine kinase (SphK), and sphingosine 1-phosphate phosphatase (SPP) are represented by their acronyms.

**Figure 2 ijms-23-08082-f002:**
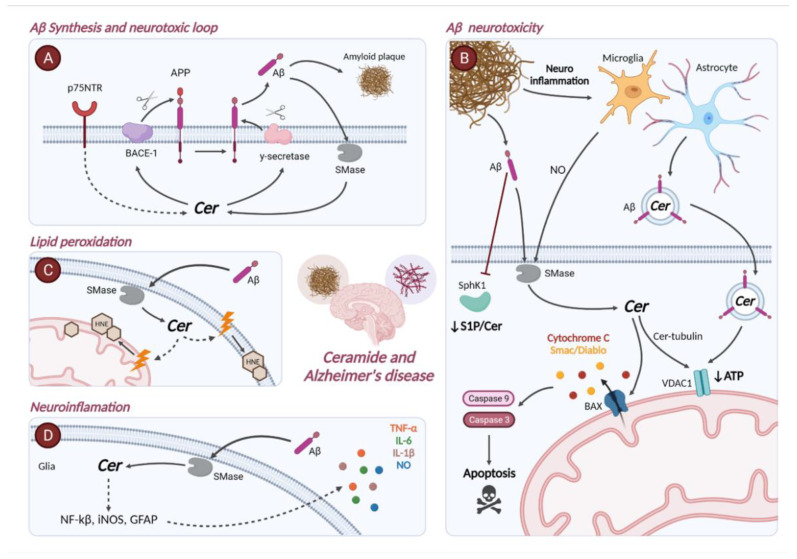
Involvement of Cer in Alzheimer’s disease. Several pieces of evidence support the involvement of Cer in the progression of AD, as it is interrelated with the formation of Aβ (**A**) and its neurotoxicity (**B**). Cer also plays a role in lipid peroxidation (**C**) and neuroinflammation (**D**) p75 neurotrophin receptor (P75NTR), β-secretase (BACE-1), amyloid precursor protein (APP), sphingomyelinase (SMase), β-amyloid (Aβ), ceramide (Cer), nitric oxide (NO), sphingosine 1-phosphate (S1P), pro-apoptotic protein Bcl-2-associated X protein (BAX), mitochondrial voltage-dependent anion channel 1 (VDAC-1), 4-Hydroxynonenal (HNE), nuclear factor kappa-light-chain-enhancer of activated B cells (NF-κB), inducible nitric oxide synthase (iNOS), glial fibrillary acidic protein (GFAP), tumor necrosis factor α (TNF-α), interleukin-6 (IL-6), and interleukin-1β (IL-1β).

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
