# Peer review of "Ceramide/Sphingosine 1-Phosphate Axis as a Key Target for Diagnosis and Treatment in Alzheimer’s Disease and Other Neurodegenerative Diseases"

_ijms, 2022, doi:10.3390/ijms23158082_

Round 1
Reviewer 1 Report
In this paper, the authors reviewed the involvement of sphingolipids (especially, ceramide and sphingosine 1-phosphate) in Alzheimer’s disease and other neurodegenerative diseases. I think the paper is intriguing and well organized, and is worth publishing in ‘International Journal of Molecular Sciences’ with minor modifications.
<specific comments>
#1. (Abstract, line 3) “high” conserved should be “highly” conserved. Please correct.
#2. (page 3, line 133) Reference#41 seems to be inappropriate for the sentence. Please check.
#3. (page 5, line 231) Reference#67 is inappropriate for the sentence. Please correct.
#4. (page 12, line 516) “oligonucleotides GW4869” should be “oligonucleotides or GW4869”. Please correct.
#5. (page 15, lines 673-676) Reference#170 seems inappropriate for the sentences. Please check.
#6. (References) Several references seem incomplete. For example, journal titles are missing in references#27, #41, and #137. And the journal year must be incorrect in reference#107. Please correct.
Author Response
We would like to thank the reviewer for the careful review. We have checked all the comments and fixed the mistakes.
Reviewer 2 Report
I have some comments on the manuscript.
1. Your manuscript should be focused on 2 AD only
2. Write an abstract concerning AD only, and also include limitations of your study at the end of the abstract section.
3. Cite PMID: 34970114, PMID: 34211570, PMID: 32973484, and PMID: 32462551 for your very first statement in the introduction section.
4. Complete editorial checking will be needed to correct the grammatical and punctuation mistakes.
5. Elaborate your major objectives, and approaches in this review manuscript in the last paragraph of the introduction section.
6. Your assessment at the end of each paragraph will be needed in the manuscript.
7. What is the impact of herbal plants and their bioactive components in the Ceramide/sphingosine 1-phosphate axis in Alzheimer’s disease?
8. Some texts are not related to their references in the manuscript. Carefully check every sentence for the same.
Author Response
We would like to thank the reviewer for the comments done.
- We are sorry but we don’t understand what the reviewer means with “2 AD only”. Our manuscript is focused on AD. However, we want to make a broad but summarized view of the involvement of sphingolipids in other neurodegenerative diseases. This point reinforces the involvement of sphingolipids in these diseases.
- We thank the reviewer for the comment. We have already written an abstract and introduction related to AD. Since it is a bibliographic review, we consider that limitations are not necessary, following the guidelines of IJMS. However, we have added some limitations to results that are discrepant from each other.
- We thank the reviewer for highlighting these interesting papers. We have added the most appropriate references to the introduction.
- We thank the reviewer for highlighting our mistakes. We have checked the paper to find and fix all of them.
- We thank the reviewer for the comment. We have already added the paragraphs.
- We have resumed papers related to SL and AD/neurodegenerative diseases. Even so, we make comments throughout the paper indicating contradictory data and/or possible unions between the works.
- We agree with the reviewer. Therefore, we have added this important information (lines 930-934)
- We thank the reviewer for highlighting our mistakes. We have checked the paper to find and fix the mistakes.
Reviewer 3 Report
In this review article Custodia et al give a comprehensive review of our current knowledge of ceramide/sphingosine 1-phosphate in neurodegenerative disease, with a special focus on Alzheimer’s disease. The article is composed in a logical order and is very thorough. This is likely to be an extremely useful reference guide to researchers into, or coming into the field, and I congratulate the authors on the it. This review deserves publication and I only have a few minor points:
Have the authors also considered including a figure or table in section 3 that summarizes the ceramide differences in each of these illnesses, and how they compare?
Figure 2 is excellent, but maybe too small. If the journal format allows it, could it be stretched to the whole page width.
I question if it is necessary to shorten ceramide to Cer in the text, as it is not an excessively long word anyway.
The authors quite often use phrases such as “as mentioned” or “aforementioned”, which I often feel are not required. If something is worth repeating then repeat it! Other than that the English and general clarity of the text is good.
Author Response
We would like to thank the reviewer for the careful review and constructive comments. We have checked all the comments and fixed the mistakes.
- We agree with the reviewer. We also think a table will be helpful to follow the paper. Therefore, as the reviewer commented, we have added a new table.
- We agree with the reviewer about Figure 2
- We appreciate the comment of the reviewer. However, as we repeatedly add different species of ceramide we think it is the best way to follow the paper. Furthermore, this shortened form is widely used in the literature.
- We agree with the reviewer. Therefore, we have modified some of these expressions.
Round 2
Reviewer 2 Report
Revised manuscript is suitable for publication.